

# *Geminal* Parahydrogen-Induced Polarization: Accumulating Long-Lived Singlet Order on Methylene Proton Pairs

Laurynas Dagys[1,*], Barbara Ripka[1,*], Markus Leutzsch[2], Gamal A. I. Moustafa[1], James Eills[3], Johannes F. P. Colell[1], and Malcolm H. Levitt[1]

[1]School of Chemistry, University of Southampton, SO17 1BJ, UK
[2]Max-Planck-Institut für Kohlenforschung, Kaiser-Wilhelm-Platz 1, D-45470 Mülheim an der Ruhr, Germany
[3]Helmholtz Institute Mainz, Johannes Gutenberg University, D-55099 Mainz, Germany
[*]These authors contributed equally to this work.

**Correspondence:** Malcolm H. Levitt (mhl@soton.ac.uk)

**Abstract.** In the majority of hydrogenative PHIP (Parahydrogen Induced Polarization) experiments, the hydrogen molecule undergoes pairwise *cis*-addition to an unsaturated precursor to occupy vicinal positions on the product molecule. However, some ruthenium-based hydrogenation catalysts induce geminal hydrogenation, leading to a reaction product in which the two hydrogen atoms are transferred to the same carbon center, forming a methylene ($CH_2$) group. The singlet order of parahydrogen

is substantially retained over the geminal hydrogenation reaction, giving rise to a singlet-hyperpolarized $CH_2$ group. Although the $T_1$ relaxation times of the methylene protons are often short, the singlet order has a long lifetime, providing that singlet-triplet mixing is suppressed, either by chemical equivalence of the protons or by applying a resonant radiofrequency field. The long lifetime of the singlet order enables the accumulation of hyperpolarization during the slow hydrogenation reaction. We introduce a kinetic model for the behaviour of the observed hyperpolarized signals, including both the chemical kinetics

and the spin dynamics of the reacting molecules. Our work demonstrates the feasibility of producing singlet-hyperpolarized methylene moieties by parahydrogen-induced polarization. This potentially extends the range of molecular agents which may be generated in a hyperpolarized state by chemical reactions of parahydrogen.



## 1 Introduction

Nuclear Magnetic Resonance (NMR) suffers from intrinsically low sensitivity, in part because of the small interaction energies between nuclear spins and magnetic fields. Hyperpolarization techniques alleviate this problem by generating nuclear spin systems with a high degree of nuclear spin polarization, enhancing the nuclear magnetization by many orders of magnitude (Ardenkjaer-Larsen et al. (2003); Maly et al. (2008); Bowers and Weitekamp (1987); Walker and Happer (1997)). Parahydrogen-induced polarization (PHIP) (Bowers and Weitekamp (1987); Natterer and Bargon (1997); Adams et al. (2009))

is a hyperpolarization method which utilizes hydrogen ($H_2$) gas enriched in the *para*-spin isomer; the enrichment is carried out by cooling $H_2$ gas over a suitable catalyst. There are two main modes of PHIP: (i) In *hydrogenative* PHIP, the strongly enhanced nuclear singlet order of *para*-enriched $H_2$ gas is substantially conserved through a pairwise catalytic transfer of the hydrogen pair onto a product molecule (Bowers and Weitekamp (1987); Natterer and Bargon (1997); Reineri et al. (2015)). The high degree of nuclear singlet order in the hydrogenation product is converted into enhanced nuclear magnetization by

symmetry-breaking nuclear spin interactions; (ii) In the SABRE (Signal Amplification By Reversible Exchange) method, reversible chemical processes are used to transfer the nuclear singlet order onto the target molecules (Adams et al. (2009); Theis et al. (2014); Truong et al. (2015); Lindale et al. (2019); Zhang et al. (2019)). PHIP has several advantages over alternative hyperpolarization techniques, such as its low cost, its compact and simple equipment requirements, and its ability to produce relatively large amounts of hyperpolarized material in a short time.

This article concerns hydrogenative PHIP experiments, which involve in most cases the *vicinal* positioning of the hydrogen substituents, i.e. the hydrogen atoms become attached to *adjacent* carbon atoms in the product molecule. Furthermore, in the case that a carbon-carbon triple bond is hydrogenated, the hydrogenation product usually has the *cis* geometry, i.e. the two hydrogens end up on the same side of the resulting double bond. This reaction specificity strongly limits the range of hyperpolarized substances accessible to hydrogenative PHIP.

Recent advances in catalytic chemistry have uncovered alternative modes of hydrogenation (Harthun et al. (1996); Leutzsch et al. (2015); Guthertz et al. (2018); Fürstner (2019)). For example, some ruthenium-based catalysts achieve *trans*-vicinal hydrogenation, meaning that the two hydrogen atoms are transferred to *opposite* sides of the resulting double bond (Leutzsch et al. (2015)). This phenomenon allows the hyperpolarization of the important metabolite fumarate in aqueous solution (Ripka et al. (2018); Eills et al. (2019)). Furthermore, under some circumstances, *geminal* hydrogenation is observed, meaning that the

two hydrogen atoms become bonded to the *same* carbon in the product molecule (Guthertz et al. (2018); Song et al. (2019)) . If *para*-enriched $H_2$ is used, the result is a methylene ($CH_2$) moiety in which the proton pair exhibits strongly enhanced nuclear singlet order, meaning a population difference between the nuclear singlet and triplet states (Carravetta et al. (2004); Carravetta and Levitt (2004); Levitt (2012); Zhang et al. (2019); Levitt (2019)). If the product molecule has sufficiently low symmetry, the $CH_2$ protons are magnetically inequivalent, allowing symmetry-breaking spin interactions to convert the nuclear singlet

order into hyperpolarized nuclear magnetization. Since methylene groups are ubiquitous in metabolites and natural products, *gem*-PHIP could potentially open up a new range of PHIP-based hyperpolarization targets.



One difficulty with *gem*-PHIP is that the associated hydrogenation reaction is usually slow (Song et al. (2019)). Furthermore, the short internuclear distance between the $CH_2$ protons leads to a strong dipole-dipole interaction, which provides an efficient $T_1$ mechanism and hence the rapid decay of hyperpolarized magnetization. The combination of a slow production rate of spin
order with a short relaxation time $T_1$ leads to weak hyperpolarization, with poor enhancement factors and low polarization levels.

Although the $T_1$ values of methylene protons are usually short, their singlet relaxation times $T_S$ can be long, exceeding 2 minutes in some cases (Carravetta and Levitt (2004)). In most cases, these long singlet lifetimes are not immediately manifest, since symmetry-breaking interactions such as chemical shift differences between the $CH_2$ protons mix the long-lived singlet
state with the rapidly relaxing triplet states. Experimental intervention is usually needed to suppress singlet-triplet mixing, either by transferring the sample to low magnetic field (Carravetta et al. (2004); Carravetta and Levitt (2005); Pileio et al. (2010); Kiryutin et al. (2019)), or by applying a resonant radiofrequency (rf) field (Carravetta and Levitt (2004); Gopalakrishnan and Bodenhausen (2006); Pileio and Levitt (2009)).

In this article we investigate the accumulation of long-lived hyperpolarized singlet order on methylene protons during a *gem*-
PHIP experiment by application of a spin-locking rf field during the slow chemical reaction (Hübler et al. (2000)). We introduce a kinetic model to describe the observed hyperpolarization levels during experiments, and provide a theoretical analysis of the spin dynamics.

**Figure 1.** Postulated mechanism for the formation of **I**. The main hydrogenation reaction leads to the product fumarate. A side reaction, involving a second acetylenedicarboxylate molecule, leads to the product **I**. The inequivalent methylene ($CH_2$) protons which derive from *para*-enriched hydrogen are shown in blue.



## 2 Geminal hydrogenation

The geminal hydrogenation reaction studied in this paper is shown in figure 1. It involves the hydrogenation of acetylenedicar-

boxylate (top left), catalyzed by the ruthenium complex $[Cp^*Ru(CH_3CN)_3]PF_6$ in $D_2O$ solution. The main product of this

reaction is the *trans*-vicinal hydrogenation product, fumarate (Ripka et al. (2018))(see Appendix B). However, in some condi-

tions, the side product **I** is also formed (the systematic name for **I**, and an NMR spectrum of the reaction mixture are given in

Supplementary Information). The side reaction is inhibited by sodium sulfite (Ripka et al. (2018)). In the current work, sodium

sulfite was not used, favouring generation of the geminal hydrogenation product **I**. The postulated reaction mechanism involves

formation of a carbene intermediate (Guthertz et al. (2018)) between the catalyst and first acetylenedicarboxylate molecule,

followed by a [3+2] cycloaddition with a second acetylenedicarboxylate molecule, dissociation of the ruthenium adduct, and

abstraction of a deuterium atom from the $D_2O$ solvent.

**I** is prone to decomposition and further reactions, and could not be isolated and subjected to standard characterisation

methods. As described in Supplementary Information the structure of **I** was verified by synthesizing a compound with the

same structure by an alternative route, followed by a comparison of the [1]H NMR spectra.

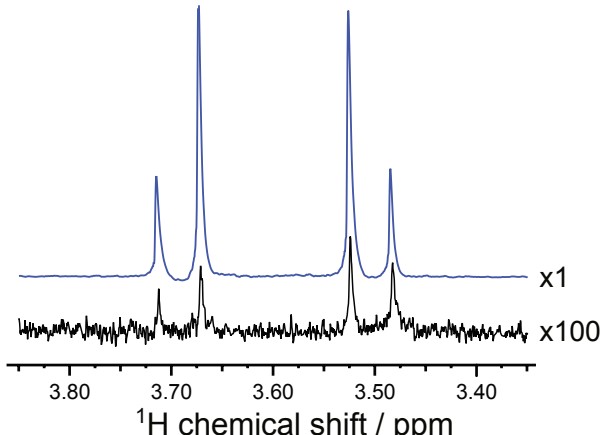

**Figure 2.** Partial [1]H NMR spectra of the reaction products at a resonance frequency of $400.0\,\mathrm{MHz}$, showing the signals from the $CH_2$ group of **I**. The hyperpolarized spectrum (blue) was acquired in a *gem*-PHIP experiment using the pulse sequence in Figure 3(a), with the intervals $\tau_1 = 90\,\mathrm{s}$ and $\tau_2 = 30\,\mathrm{s}$. The spectrum in black was obtained by waiting 90 seconds after the conclusion of the experiment, and taking the Fourier transform of the NMR signal induced by a single $\pi/2$ pulse. The spectrum shows an AB peak pattern, with intensity distortions from residual hyperpolarization. The AB spectrum is consistent with a chemical shift difference of $\Delta\delta = 0.197\,\mathrm{ppm}$ and a two-bond J-coupling $|^2J| = 16.8\,\mathrm{Hz}$. The signal enhancement factor in the *gem*-PHIP experiment is estimated to be $\sim 300$, which corresponds to a [1]H polarization level of $\sim 0.9\,\%$.





This paper focusses on the two $CH_2$ protons of the product molecule **I** which derive from the *para*-enriched $H_2$ reagent. This proton pair is highlighted in blue in figure 1. The chemical equivalence of these $CH_2$ protons is broken by the chiral centre four bonds away, on the opposite side of the 5-membered ring.

Figure 2 shows the $CH_2$ region of the $^1H$ NMR spectrum of the reaction product. The black spectrum is the Fourier transform of the NMR signal induced by a single $\pi/2$ pulse, obtained 90 seconds after the conclusion of the chemical reaction with *para*-enriched hydrogen. It displays a typical AB pattern, albeit with some spectral intensity distortions from residual hyperpolarization effects (see Appendix B for further explanation). The two protons have a chemical shift difference of $\Delta\delta = 0.197$ ppm and a two-bond J-coupling of $|^2J| = 16.8$ Hz.

The nuclear spin relaxation characteristics of **I** were estimated at room temperature (295 K) and a magnetic field of 9.4 T, using standard techniques (see Supplementary Information). The spin-lattice relaxation time of the $CH_2$ protons is given by $T_1 = 1.23 \pm 0.14$ s. The singlet relaxation time of the $CH_2$ protons under the same conditions is $T_S = 61.1 \pm 7.1$ s. Unfortunately, the chemical instability of **I** made it impossible to estimate the relaxation times under the much warmer conditions of the *gem*-PHIP reaction.

## 3 Results

### 3.1 *gem*-PHIP

Parahydrogen-induced hyperpolarization of **I** was demonstrated using the pulse sequence in figure 3(a). Bubbling of *para*-enriched hydrogen was conducted for an interval $\tau_1 = 90$ s in the presence of a radiofrequency spin-locking field (Hübler et al. (2000)), whose frequency corresponds to the mean chemical shift of the $CH_2$ protons. The spin-locking field amplitude corresponded to a $^1H$ nutation frequency of $\omega_{\mathrm{nut}}/2\pi = 1.0$ kHz. Bubbling was switched off and the spin-locking continued for a further interval of $\tau_2 = 30$ s. This gave time for the bubbles to dissipate and for hyperpolarized singlet order to accumulate during the on-going hydrogenation reaction.

Hyperpolarized singlet order was converted into in-phase magnetization by the sequence of three delays and two radiofrequency pulses shown in figure 3. This sequence converts magnetization into singlet order in weakly coupled spin-1/2 pairs (Sarkar et al. (2007)). The ideal values of the pulse sequence delays, in the case of infinitely short pulses, are $\tau_3 = |\pi/\omega_\Delta|$ and $\tau_4 = 1/(4J)$, where the chemical shift frequency difference is $\omega_\Delta = \omega^0 \Delta\delta$ and $\omega^0$ is the Larmor frequency. In practice, the following pulse sequence intervals were used: $\tau_3 = 6.49$ ms and $\tau_4 = 14.97$ ms.

Figure 2 shows the $^1H$ NMR spectrum of **I**, hyperpolarized by *gem*-PHIP (blue spectrum). The NMR signals of the $CH_2$ protons are enhanced by a factor of $\sim 300$ as compared to the spectrum taken 90 s after the end of the pulse sequence (black spectrum). This enhancement factor corresponds to a modest polarization level of $\sim 0.9\%$. Although the achieved polarization level is not spectacular this experiment demonstrates the feasibility of the *gem*-PHIP of methylene protons, providing that a spin-locking field is used to stabilize the hyperpolarized singlet order.





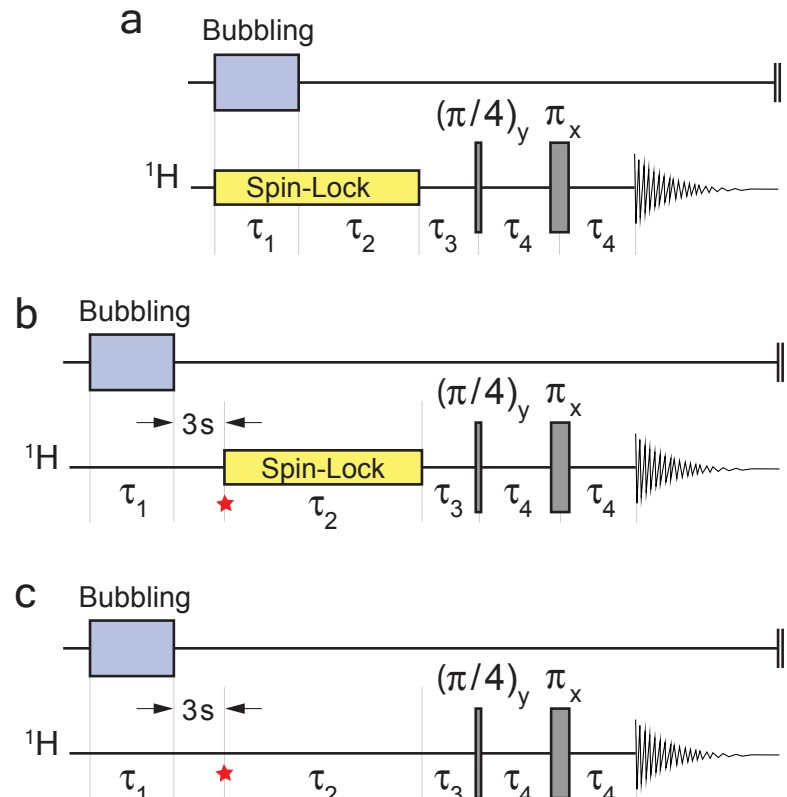

**Figure 3.** Experimental timing sequences. **(a)** Procedure for the demonstration of *gem*-PHIP. Bubbling of *para*-enriched $H_2$ is conducted for an interval $\tau_1$ in the presence of a radio-frequency spin-locking field in order to suppress singlet-triplet mixing in the reaction product **I**. The spin-locking continues for a further interval $\tau_2$, followed by a two-pulse sequence to convert the hyperpolarized singlet order to in-phase magnetization (Sarkar et al. (2007)). The experimental delays were $\tau_3 = 6.49$ ms and $\tau_4 = 14.97$ ms. **(b)** Procedure for demonstrating the accumulation of singlet order during spin-locking. The spin-lock field is applied during the variable interval $\tau_2$, with an amplitude corresponded to a nutation frequency $\omega_{\mathrm{nut}}/(2\pi) = 1.0\,\mathrm{kHz}$. The star symbol refers to the time point discussed in the text. **(c)** The same sequence as for (b), but without spin-locking during the variable $\tau_2$ interval. The interval $\tau_1$ was set to 90 s for (a) and 17 s for (b) and (c).

## 3.2 Hyperpolarization decay

Figure 4 shows the dependence of the integrated *gem*-PHIP signal intensity on the spin-locking interval $\tau_2$ in figure 3(a), with the bubbling time $\tau_1$ increased to 90 s. Each point in figure 4 is the result of an independent experiment, performed on a fresh aliquot of the stock solution, with the signal amplitude normalized against the integrated amplitude of the thermal equilibrium spectrum, obtained 90 s after the pulse sequence has finished. The integrated signal amplitude follows a monoexponential decay function with a time constant of $151 \pm 9$ s. As discussed below, this time constant may be assigned to the decay time constant $T_S$ for singlet order in the presence of the spin-locking field.

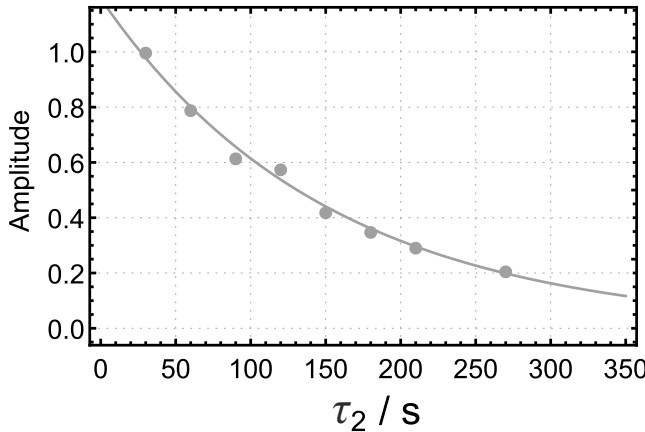

**Figure 4.** Dependence of the integrated *gem*-PHIP signal amplitude for the $CH_2$ protons of **I** on the interval $\tau_2$ in the pulse sequence of figure 3(a), with $\tau_1$ fixed to $90\,\text{s}$. Solid line: fit to equation 19 with $f_a A_a = 1.79$, $f_a B_a = 0$ and time constant $T_S^{\mathbf{I}} = 151$ s for singlet order decay.

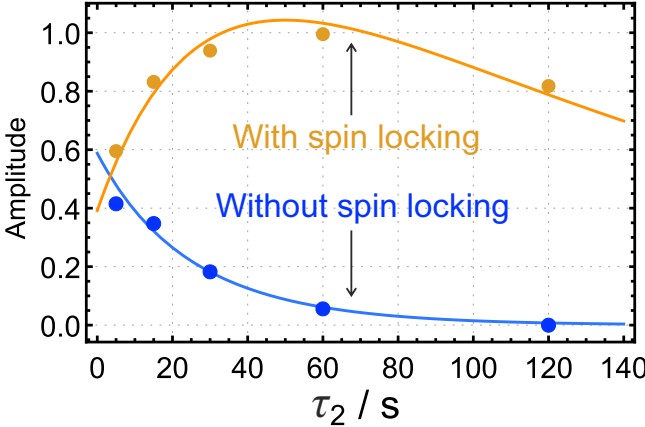

**Figure 5.** Dependence of the integrated *gem*-PHIP signal amplitudes of the $CH_2$ protons of **I** on the interval $\tau_2$ in the pulse sequences of figure 3(b) and 3(c), with $\tau_1$ fixed to $17\,\text{s}$. The orange symbols show the amplitudes for the case of a spin-locking field during the $\tau_2$ interval (sequence in figure 3(b)), with an amplitude corresponding to the nutation frequency $\omega_{\text{nut}}/2\pi = 1.0\,\text{kHz}$. The blue symbols show the amplitudes for experiments without a spin-locking field during the $\tau_2$ interval (sequence in figure 3(c)). The orange and blue solid lines show the functions $a_b(\tau_2)$ and $a_c(\tau_2)$ (equations 19 and 23 respectively), with the parameters $T_S^{\mathbf{I}} = 151$ s; $T_\Sigma^{\text{H}2} = 28.7$ s; $T_{\text{zz}}^{\mathbf{I}} = 13.2$ s; $f_b C_{\text{SO}}^{\text{H}2}(0) \times k = 0.059\,\text{s}^{-1}$; $f_c C_{\text{zz}}^{\mathbf{I}}(0) = -1.2$; $f_b = f_c$.



### 3.3 Accumulation of hyperpolarized singlet order

The pulse sequence in figure 3(b) was used to study the accumulation of hyperpolarized singlet order on the $CH_2$ protons of **I** during the slow geminal hydrogenation reaction. The experiment started by bubbling *para*-enriched $H_2$ gas through the NMR tube for $\tau_1 = 17\,\mathrm{s}$, in order to saturate the solution. The sample was then allowed to rest for a settling time of 3 s in order to dissipate bubbles and to achieve good sample and field homogeneity. The trajectory of the hyperpolarized spin order during the subsequent interval was followed by varying the interval $\tau_2$ in a series of independent experiments, each one performed

on a separate aliquot of the same stock solution. Experiments were also performed without spin-locking during the $\tau_2$ interval (figure 3c).

The results of this investigation are shown in figure 5. When a spin-locking field is applied during the $\tau_2$ interval (figure 3b), the hyperpolarized signals first increase and then decay (orange symbols). If no spin-locking field is applied during the $\tau_2$ interval (figure 3c), the hyperpolarized NMR signals decay monotonically with respect to $\tau_2$ (blue symbols).

## 4 Kinetic Analysis

Figure 6 shows the simplified kinetic model which is used to interpret these results. The dynamics of the system may be analyzed in terms of the chemical kinetics of the hydrogenation reaction as well as the spin dynamics of the product molecule **I**. Although the chemical kinetics depend only on concentrations and physical conditions, the spin dynamical pathways may be manipulated experimentally with fine time resolution, for example by turning spin-locking fields on and off. The experimental

results derive from an interplay between the chemical and spin-dynamical domains. Similar analyses have been performed in different contexts (Kaptein (1972); Hübler et al. (1999); Goez (2009); Pravdivtsev et al. (2015); Emondts et al. (2017); Lindale et al. (2019); Barskiy et al. (2019)).

### 4.1 Chemical kinetics

After *para*-enriched $H_2$ gas is introduced into solution by bubbling, it starts to react with the acetylenedicarboxylate precursor,

catalyzed by the ruthenium complex. As depicted in figure 1, this is a complex process with the generation of several products, and with the production of **I** requiring an additional precursor molecule. Nevertheless, for the sake of simplicity, and since the acetylenedicarboxylate precursor is in excess, the reaction leading to **I** is assumed to be first-order with respect to the *para*-$H_2$ reagent and to proceed with rate constant $k$.

After the bubbling has stopped, the concentrations of the $H_2$ reagent and the product molecule **I** are assumed to follow the

simple kinetic equations:

$$\frac{\mathrm{d}}{\mathrm{d}t}\left[H_2\right]_t = -k_{\mathrm{tot}}\left[H_2\right]_t$$
$$\frac{\mathrm{d}}{\mathrm{d}t}\left[\mathbf{I}\right]_t = +k\left[H_2\right]_t \tag{1}$$





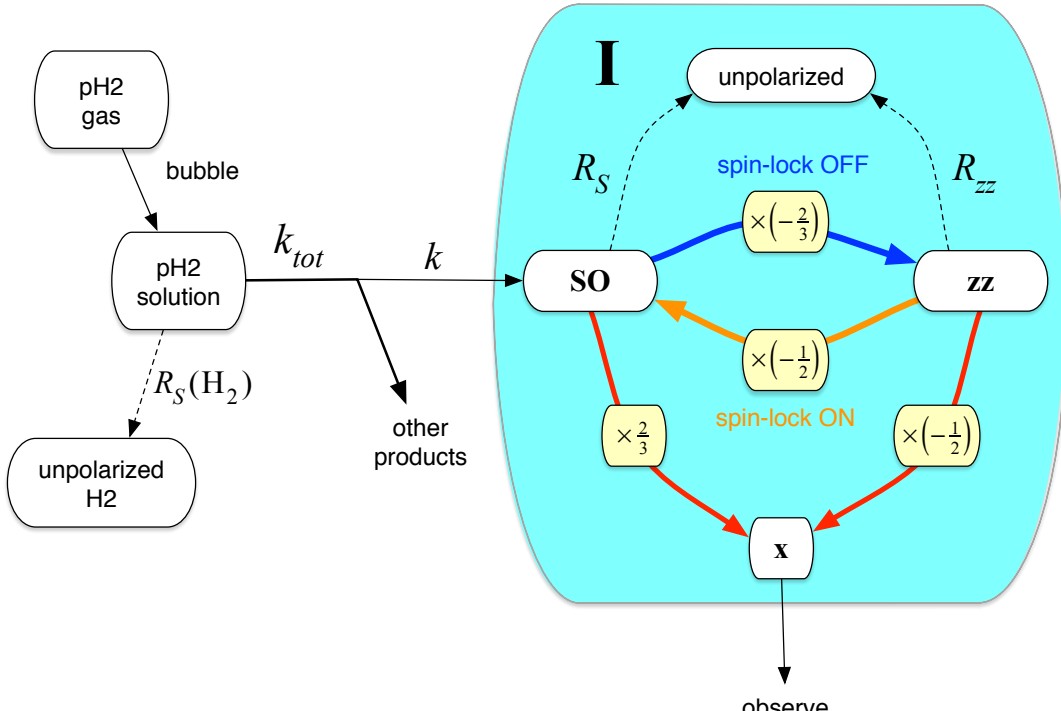

**Figure 6.** Kinetic model for *gem*-PHIP. The chemical reaction of *para*-enriched $H_2$ with the acetylenedicarboxylate precursor and ruthenium-based catalyst (not shown) proceeds with rate constant $k$. The shaded area includes the product molecule **I** in different spin polarization states: unpolarized (top), in a state of singlet nuclear spin order (left), in a state of zz-order (right), and with observable x-magnetization (bottom). Singlet order decays with rate constant $R_S = T_S^{-1}$; zz-order decays with rate constant $R_{zz} = T_{zz}^{-1}$. If no spin-locking field is present, singlet order is rapidly converted into zz-order, with a conversion factor of $-2/3$ (blue arrow and yellow box). If a spin-locking field is applied, zz-order is instantaneously projected onto singlet order, with a conversion factor of $-1/2$ (orange arrow and yellow box). Singlet order and zz-order may be converted into observable $x$-magnetization by the two-pulse sequence in figure 3 (red arrows and yellow boxes). The conversion factors in this case are $2/3$ for singlet order and $-1/2$ for zz-order.

where $k_{tot}$ is the rate constant for all hydrogenation reactions, including those that do not lead to the product molecule **I**, $k_{tot} > k$. The differential equations 1 are easily solved to show that the solution concentration of $H_2$ decays exponentially with time, while the concentration of the product molecule **I** increases:

$$\left[H_2\right]_t = \left[H_2\right]_0 \exp(-k_{tot}t) \tag{2}$$

$$\left[\mathbf{I}\right]_t = \left[\mathbf{I}\right]_\infty (1 - \exp(-kt)) \tag{3}$$

where the limiting value of the concentration of **I** is given by

$$\left[\mathbf{I}\right]_\infty = \frac{k}{k_{tot}}\left[H_2\right]_0 \tag{4}$$



As expected, the limiting yield of $\mathbf{I}$ depends on the ratio of the productive rate constant $k$ to the total rate constant of all
hydrogenation reactions $k_{\text{tot}}$.

## 4.2 Spin Dynamics

In this Section we discuss some general spin dynamical scenarios relevant to parahydrogen-enhanced NMR experiments, and
in Section 4.3 we evaluate the spin dynamics and chemical kinetics specific to the experiments performed in this work. The
spin dynamics in this section were evaluated with the assistance of *SpinDynamica* software (Bengs and Levitt (2018)).

### 4.2.1 Singlet Order

The proton spins of *para*-enriched $H_2$ are in a state of enhanced nuclear singlet order, described as the difference between the
population of the nuclear singlet state and the mean population of the nuclear triplet states:

$$\mathbf{SO} = \langle S_0 | \rho | S_0 \rangle - \frac{1}{3} \sum_{M=-1}^{+1} \langle T_M | \rho | T_M \rangle \tag{5}$$

where $\rho$ is the spin density operator, and the singlet and triplet states are defined in terms of the Zeeman product states as
follows (Levitt (2012)):

$$
\begin{aligned}
|S_0\rangle &= 2^{-1/2}(|\alpha\beta\rangle - |\beta\alpha\rangle) \\
|T_{+1}\rangle &= |\alpha\alpha\rangle \\
|T_0\rangle &= 2^{-1/2}(|\alpha\beta\rangle + |\beta\alpha\rangle) \\
|T_{-1}\rangle &= |\beta\beta\rangle
\end{aligned}
\tag{6}
$$

Singlet order $\mathbf{SO}$ may be regarded as the expectation value of the singlet order operator $Q_{\text{SO}}$, which is defined as follows:

$$Q_{\text{SO}} = |S_0\rangle\langle S_0| - \frac{1}{3} \sum_{M=-1}^{+1} |T_M\rangle\langle T_M| = -\frac{4}{3}\mathbf{I}_1 \cdot \mathbf{I}_2 \tag{7}$$

so that

$$\mathbf{SO} = \langle Q_{\text{SO}} \rangle = \text{Tr}\{Q_{\text{SO}}^\dagger \rho\} \tag{8}$$

$H_2$ gas in thermal equilibrium at room temperature, with an *ortho*:*para* ratio very close to 3:1, has negligible singlet order,
$\mathbf{SO} \simeq 0$. Pure *para*-$H_2$ gas has singlet order $\mathbf{SO} = 1$. The current work employs $H_2$ gas which is enriched with the *para* spin
isomer by thermal equilibration at 77 K. This yields an *ortho*:*para* ratio of approximately 1:1, corresponding to a singlet order
of $\mathbf{SO} \simeq 1/3$. Assuming that the nuclear spin states are substantially unchanged through the chemical reaction, the product
molecule $\mathbf{I}$ is formed with its methylene protons in a similar state of finite singlet order, $\mathbf{SO} \simeq 1/3$.

The singlet order operator $Q_{\text{SO}}$ is an exact eigenoperator of the spin propagation superoperator in the case of a magnetically
equivalent spin-pair system such as for $H_2$ gas. However, in the product molecule $\mathbf{I}$, the chiral centre breaks the equivalence of





the methylene protons, so that the operator $Q_{\mathrm{SO}}$ is no longer an eigenoperator of the evolution. The chemical shift difference induces singlet-triplet transitions which mix the operator $Q_{\mathrm{SO}}$ with other operators. However, if a sufficiently strong spin-locking field is applied, the singlet-triplet transitions are suppressed, so that the order **SO** is substantially unchanged during the evolution, except for a decay due to relaxation processes (Pileio and Levitt (2009)). The decay rate constant is given

by $R_{\mathrm{S}} = T_{\mathrm{S}}^{-1}$, where $T_{\mathrm{S}}$ is the time constant for singlet order decay, and which is often much longer than the relaxation time constant $T_1$ for longitudinal magnetization. The decay of singlet order in the presence of a spin-locking field, with rate constant $R_{\mathrm{S}}$, is shown in figure 6 by the dashed arrow running upwards, connecting the **SO** state of the reaction product **I** to the unpolarized state.

### 4.2.2   zz-Order

A different type of nuclear spin order is called zz-order (Sørensen et al. (1984)), and corresponds to the expectation value of an operator $Q_{\mathrm{zz}}$, defined as follows:

$$Q_{\mathrm{zz}} = 2 I_{1z} I_{2z}$$

$$\mathbf{zz} = \langle Q_{\mathrm{zz}} \rangle = \mathrm{Tr}\{Q_{\mathrm{zz}}^{\dagger} \rho\}$$

(9)

 In the absence of a spin-locking field, and if there is a relatively large chemical shift difference between the coupled spins, the operator $Q_{\mathrm{zz}}$ is a better approximation to an eigenoperator of the spin evolution propagator than the singlet order operator

$Q_{\mathrm{SO}}$. The relaxation of the system can be complex and multi-exponential in this case. Nevertheless, for the sake of simplicity, we assume here a single rate constant $R_{\mathrm{zz}} = T_{\mathrm{zz}}^{-1}$ for the zz-order in the absence of a spin-locking field. The time constant $T_{\mathrm{zz}}$ is expected to be close to the ordinary spin-lattice relaxation time constant $T_1$. The decay of zz-order in the absence of a spin-locking field, with rate constant $R_{\mathrm{zz}}$, is shown in figure 6 by the dashed arrow running upwards connecting the **zz** state of the reaction product **I** to the unpolarized state.

### 195   4.2.3   Spin-locking OFF

Suppose that the molecules of **I** are in a state of enhanced singlet order **SO**. This state is stable if a spin-locking field is continuously applied, and decays monotonically with the time constant $T_{\mathrm{S}}$, However, if the spin-locking field is turned off, the chemical shift difference between the methylene protons leads to rapid singlet-triplet mixing. The zz-order operator $Q_{\mathrm{zz}}$ is an approximate eigenoperator of the evolution in this case, instead of the singlet-order operator $Q_{\mathrm{SO}}$. Hence, any singlet order **SO**

which is present when the spin-locking field is turned off is projected onto the zz-order operator $Q_{\mathrm{zz}}$. The remaining spin order corresponds to zero-quantum coherences which rapidly oscillate and decay. These additional components may be ignored to a good approximation, providing that the spin-locking field remains turned off for an interval long compared to the difference in chemical shift frequencies.

The zz-order created by this projection process is given by

$$\mathbf{zz} = \frac{\mathrm{Tr}\{Q_{\mathrm{zz}}^{\dagger} Q_{\mathrm{SO}}\}}{\mathrm{Tr}\{Q_{\mathrm{zz}}^{\dagger} Q_{\mathrm{zz}}\}} \mathbf{SO} = -\frac{2}{3} \mathbf{SO}$$

(10)





The projection of **SO** onto **zz** is depicted by the blue arrow in figure 6, annotated by the projection factor $-2/3$ (yellow box).

### 4.2.4 Spin-locking ON

Suppose that the molecules of **I** are in a state of enhanced zz-order **zz**. The corresponding operator $Q_{zz}$ is an eigenoperator of the spin evolution in the absence of a spin-locking field. However, if the spin-locking field is turned on, singlet-triplet mixing

is suppressed, and the zz-order operator $Q_{zz}$ is no longer an eigenoperator of the spin evolution. Any zz-order which is present when the spin-locking field is turned on is projected onto the singlet order operator $Q_{SO}$. The remaining spin order corresponds to high-rank spin order terms which rapidly dephase under radiofrequency field inhomogeneity.

The singlet order created by this projection process is given by

$$\mathbf{SO} = \frac{\mathrm{Tr}\{Q_{SO}^{\dagger}Q_{zz}\}}{\mathrm{Tr}\{Q_{SO}^{\dagger}Q_{SO}\}}\mathbf{zz} = -\frac{1}{2}\mathbf{zz} \tag{11}$$

The projection of **zz** onto **SO** is depicted by the orange arrow in figure 6, annotated by the projection factor $-1/2$ (yellow box).

### 4.2.5 Signal read-out

The spin orders **zz** and **SO** are observed by applying the two-pulse sequence given in figure 3(c), and described in Sarkar et al. (2007). This sequence converts both types of spin order into observable transverse magnetization, which induces a time-domain

NMR signal in the subsequent interval of free precession. The read-out transformations may be written as follows:

$$UQ_{SO}U^{\dagger} = a(\mathbf{SO} \to \mathbf{x})I_x + \dots$$
$$UQ_{zz}U^{\dagger} = a(\mathbf{zz} \to \mathbf{x})I_x + \dots \tag{12}$$

where $U$ is the propagator for the two-pulse sequence and the dots denote operators which are orthogonal to $I_x$. These amplitudes may be calculated as follows:

$$a(\mathbf{SO} \to \mathbf{x}) = \frac{\mathrm{Tr}\{I_x^{\dagger}UQ_{SO}U^{\dagger}\}}{\mathrm{Tr}\{I_x^2\}}$$
$$a(\mathbf{zz} \to \mathbf{x}) = \frac{\mathrm{Tr}\{I_x^{\dagger}UQ_{zz}U^{\dagger}\}}{\mathrm{Tr}\{I_x^2\}} \tag{13}$$

In an ideal weakly-coupled spin system, with infinitely short, ideal, radiofrequency pulses, and delays given by $\tau_3 = |\pi/\omega_\Delta|$ and $\tau_4 = 1/(4J)$, the transformation amplitudes are as follows:

$$a(\mathbf{SO} \to \mathbf{x}) = \frac{2}{3}$$
$$a(\mathbf{zz} \to \mathbf{x}) = -\frac{1}{2} \tag{14}$$

These transformations are indicated by the red arrows and yellow boxes in figure 6.

The integrated amplitude of the NMR spectrum obtained by Fourier transformation of the NMR signal is therefore propor-

tional to the **zz** and **SO** orders before the read-out sequence is applied, multiplied by the transformation factors in equation 14.



### 4.3 Analysis of experimental trajectories

The chemical kinetics and spin dynamics may be combined to achieve an understanding of the trajectories in figures 4 and 5, generated by the timing sequences shown in figure 3.

#### 4.3.1 Trajectories with Spin Locking

The pulse sequences in figures 3(a,b) both examine the dependence of hyperpolarized signals on the duration $\tau_2$ of a spin-locking interval. However the state of the spin system at the start of the $\tau_2$ interval is different in the two procedures. In figure 3(a), which provides the results shown in figure 4, spin locking is applied continuously during the bubbling interval and continued during the variable delay $\tau_2$. In the sequence of figure 3(b), on the other hand, which provides the orange data points in figure 5, the spin locking is interrupted for 3 s before the $\tau_2$ interval starts.

In both cases, the evolution of the singlet order during the spin-locking interval obeys the following differential equations:

$$\frac{\mathrm{d}}{\mathrm{d}t} C_{\mathrm{SO}}^{\mathrm{H}_2}(t) = -R_\Sigma^{\mathrm{H}_2} C_{\mathrm{SO}}^{\mathrm{H}_2}(t)$$

$$\frac{\mathrm{d}}{\mathrm{d}t} C_{\mathrm{SO}}^{\mathbf{I}}(t) = +k C_{\mathrm{SO}}^{\mathrm{H}_2}(t) - R_S^{\mathbf{I}} C_{\mathrm{SO}}^{\mathbf{I}}(t) \tag{15}$$

The notation $C_{\mathrm{SO}}^{X}(t)$ indicates the total amplitude of singlet spin order for the species $X$ at time point $t$, taking into account the concentration of $X$ as well as its spin state. The decay rate constant for singlet order in compound $\mathbf{I}$, due to spin-dynamical

processes, is denoted $R_S^{\mathbf{I}} = T_S(\mathbf{I})^{-1}$. The total decay rate constant for $H_2$ singlet order, due to the combination of chemical and spin-dynamical processes, is denoted

$$R_\Sigma^{\mathrm{H}_2} = k_{\mathrm{tot}} + R_S^{\mathrm{H}_2} , \tag{16}$$

where $R_S^{\mathrm{H}_2}$ denotes the decay rate constant for $H_2$ singlet order, due to *para*-to-*ortho* conversion in solution, in the presence of the hydrogenation catalyst but in the absence of a hydrogenation reaction. Note that this rate constant may be greatly increased

by the presence of the catalyst, since transient binding of $H_2$ molecules with the catalyst provides an efficient mechanism for *ortho-para* conversion.

   Equations 15 may be solved to obtain the following trajectory of the singlet order for compound $\mathbf{I}$ under spin-locking:

$$C_{\mathrm{SO}}^{\mathbf{I}}(\tau_2, i) = A_i \exp\{-R_S^{\mathbf{I}} \tau_2\} + B_i \exp\{-R_\Sigma^{\mathrm{H}_2} \tau_2\} \tag{17}$$

where the coefficients are

$A_i = C_{\mathrm{SO}}^{\mathbf{I}}(0, i) - B_i$

$$B_i = \frac{k C_{\mathrm{SO}}^{\mathrm{H}_2}(0, i)}{R_S^{\mathbf{I}} - R_\Sigma^{\mathrm{H}_2}} \tag{18}$$

The index $i$ refers to the two first pulse sequences in figure 3, $i \in \{a, b\}$. The symbol $C_{\mathrm{SO}}^{\mathbf{I}}(0, i)$ is the total amplitude of $H_2$ singlet order at the start of the spin-lock interval in experiment $i$, taking into account the concentration of $\mathbf{I}$ as well as its spin state.





The amplitude factor for the read-out of singlet order is given by $(+2/3)$, as shown by equation 14. Hence the integrated signal amplitudes for the sequences in figure 3(a,b) are given by:

$$a_i(\tau_2) = \frac{2}{3} f_i \left( A_i \exp\{-R_S^{\mathbf{I}} \tau_2\} + B_i \exp\{-R_\Sigma^{\mathrm{H}_2} \tau_2\} \right) \qquad (19)$$

where $f_i$ are instrumental factors and $i \in \{a, b\}$. The signal trajectories have a biexponential form, in general.

### 4.3.2   Trajectory without Spin Locking

The sequence in figure 3(c) is identical to that in figure 3b, except for the absence of the spin-locking field during the $\tau_2$ interval. In the absence of spin locking, the relevant eigenoperator of the spin evolution during the $\tau_2$ interval is the zz-operator $Q_{\mathrm{zz}}$ (equation 9). The combined chemical/spin dynamics of the system is described by the following differential equations:

$$\frac{\mathrm{d}}{\mathrm{d}t} C_{\mathrm{SO}}^{\mathrm{H}_2}(t) = -R_\Sigma^{\mathrm{H}_2} C_{\mathrm{SO}}^{\mathrm{H}_2}(t)$$
$$\frac{\mathrm{d}}{\mathrm{d}t} C_{\mathrm{zz}}^{\mathbf{I}}(t) = (-\frac{2}{3}) k C_{\mathrm{SO}}^{\mathrm{H}_2}(t) - R_{\mathrm{zz}}^{\mathbf{I}} C_{\mathrm{zz}}^{\mathbf{I}}(t) \qquad (20)$$

The factor $(-2/3)$ appears since the $\mathrm{H}_2$ singlet order is projected onto the zz-order of the product molecule $\mathbf{I}$ upon hydrogenation, as described in section 4.2.3.

   The differential equations 20 may be solved to obtain the following trajectory for the zz-order of compound $\mathbf{I}$, under the pulse sequence of figure 3(c):

$$C_{\mathrm{zz}}^{\mathbf{I}}(t, c) = A_c \exp\{-R_{\mathrm{zz}}^{\mathbf{I}} t\} + B_c \exp\{-R_\Sigma^{\mathrm{H}_2} t\} \qquad (21)$$

where the coefficients are

$$A_c = C_{\mathrm{zz}}^{\mathbf{I}}(0, c) - B_c$$
$$B_c = -\frac{2k C_{\mathrm{SO}}^{\mathrm{H}_2}(0, c)}{3 \left( R_{\mathrm{zz}}^{\mathbf{I}} - R_\Sigma^{\mathrm{H}_2} \right)} \qquad (22)$$

Here $C_{\mathrm{zz}}^{\mathbf{I}}(0, c)$ is the zz-order of compound $\mathbf{I}$ at the beginning of the $\tau_2$ interval. The zz-order at the end of the $\tau_2$ interval is transformed into observable x-magnetization by applying a sequence of two pulses and three delays. The amplitude factor for

the read-out of zz-order is given by $(-1/2)$, as shown by equation 14. Hence the integrated signal amplitude for the sequence in figure 3(c) is proportional to:

$$a_c(\tau_2) = -\frac{1}{2} f_c \left( A_c \exp\{-R_{\mathrm{zz}}^{\mathbf{I}} \tau_2\} + B_c \exp\{-R_\Sigma^{\mathrm{H}_2} \tau_2\} \right) \qquad (23)$$

This also has the form of a bi-exponential decay.

   Since the sequences in figure 3(b) and 3(c) are the same up to the start of the $\tau_2$ interval (indicated by the star in the pulse

sequence diagrams), the instrumental factors are identical ($f_b = f_c$) and we can write

$$C_{\mathrm{SO}}^{\mathbf{I}}(0, b) = -\frac{1}{2} C_{\mathrm{zz}}^{\mathbf{I}}(0, c) , \qquad (24)$$





using the projection in equation 11. Hence the signal amplitudes for these two experiments have the following relationship at the start of the $\tau_2$ interval:

$$\frac{\alpha_b(0)}{\alpha_c(0)} = \frac{2}{3} \tag{25}$$

The difference in starting points is evident in the theoretical curves shown by the solid lines in figure 5.

### 4.3.3  Data fitting

The data sets of figures 4 and 5 were fitted simultaneously using the set of global parameters. All three data sets were well fitted by the functions $a_a(\tau_2)$, $a_b(\tau_2)$ and $a_c(\tau_2)$ (equations 19 and 23) with the following parameters: $T_S^{\mathbf{I}} = 151 \pm 9$ s; $T_\Sigma^{H_2} = 28.7 \pm 3.8$ s; $T_{zz}^{\mathbf{I}} = 13.2 \pm 1.3$ s; $f_a A_a = 1.79 \pm 0.07$; $f_b B_a \approx 0$; $f_b C_{SO}^{H_2}(0,b) \times k = 0.059 \pm 0.007$ s$^{-1}$; $f_c C_{zz}^{\mathbf{I}}(0,c) = -1.2 \pm 0.1$.

All rate constants are expressed here as time constants, i.e. $T_X = R_X^{-1}$. The parameters $f_b C_{SO}^{H_2}(0,b)$ and $k$ interact strongly in the fit and could not be determined independently.

For these parameters, the trajectory in figure 4 is very close to a single-exponential decay with time constant $T_S^{\mathbf{I}} = 151 \pm 9$ s. For the case of the orange curve in figure 5, on the other hand, the singlet order on $\mathbf{I}$ starts at a relatively low level. The long singlet decay time constant allows accumulation of singlet order as the reaction proceeds in the presence of the spin-locking

field. This accumulation gives rise to the rising initial trajectory of the orange curve in figure 5. The comparatively short time constant for the decay of zz-order, $T_{zz}^{\mathbf{I}} \simeq 13.2$ s, allows no time for zz-order to accumulate in the absence of a spin-locking field, giving rise to the monotonically decaying blue curve in figure 5.

The singlet decay time constant for the methylene protons of compound $\mathbf{I}$ was determined independently by non-hyperpolarized experiments (see Supplementary Information). These experiments were performed at a much lower sample temperature

of 295 K to avoid the decomposition of $\mathbf{I}$. The estimated value of $T_S^{\mathbf{I}}$ at 295 K and a magnetic field of 9.41 T is $61.1 \pm 7.1$ s. This value is much shorter than the estimate of $T_S^{\mathbf{I}} = 151 \pm 9$ s from the hyperpolarization trajectories. The discrepancy may be due in part to a reduction in rotational correlation time for the molecules of $\mathbf{I}$ at the elevated temperature used in the PHIP experiments.

## 5  Materials and Methods

All experiments were conducted on a Bruker Avance Neo 400 MHz (9.41 T) system equipped with a 5 mm BBO probe. The excitation pulses were applied on-resonance with the doublet at 3.6 to 3.7 ppm. Their amplitude corresponded to a nutation frequency of $\sim$20 kHz. The spectral width was set to 20 ppm with sampling of 65k points.

The reagent solution consisted of 100 mM disodium acetylenedicarboxylate and 6 mM [Cp*Ru(CH$_3$CN)$_3$]PF$_6$ dissolved in D$_2$O. All sample solutions were prepared by mixing the components, sonicating the mixture for 5 min at 50°C and filtering

it through a 0.2 $\mu$m pore-size syringe filter with a nylon membrane.





*Para*-enriched hydrogen was produced by slowly passing hydrogen gas through an iron oxide catalyst submerged in liquid nitrogen to obtain 50% *para*-enriched hydrogen. A container was pressurized with 10 bar of *para*-enriched $H_2$ to contain gas for a whole series of experiments at 4 bar of parahydrogen pressure.

Hydrogenation experiments of disodium acetylenedicarboxylate and catalyst $[Cp^*Ru(CH_3CN)_3]PF_6$ were carried out strictly according to the experimental procedure in Table 1. For each experiment, a 300 $\mu$L aliquot was used from a stock solution. Bubbling was performed in a 5 mm Wilmad® quick pressure valve NMR tube through a 1/16" PEEK capillary, using 4 bar parahydrogen pressure, 60 °C (333 K) temperature and a gas flow of 400 sccm.

Spin-locking was performed by irradiating a continuous wave rf field at the mean resonance frequency of the $CH_2$ protons and with an amplitude corresponding to a 1.0 kHz nutation frequency.

| Duration | Event |
|---|---|
| - | Inject 300 $\mu$L of sample solution into the NMR tube |
| 1 min | Pressurise and bubble sample with inert gases at 4 bar. Depressurize |
| 10 min | Put sample in the magnet and raise temperature from 40°C to 60°C |
| 10 s | Pressurise sample with parahydrogen |
| 10 s | Bubble sample with parahydrogen to saturate sample and to pre-activate the catalyst |
| 5 min | Establish field homogeneity (shimming) |
| Variable | Perform the experiment |
| - | Lower temperature to 40°C and depressurize the sample |

**Table 1.** Experimental procedure for gem-PHIP experiments.

**6 Conclusions**

In this work we have demonstrated *geminal* hydrogenation of a precursor molecule using *para*-enriched hydrogen gas. We show that singlet order for the methylene proton pair may be maintained by application of a spin-locking field, and that the proton singlet order in the product molecule relaxes with the time constant $T_S^{I} \simeq 151\,\mathrm{s}$, which is more than 50 times $T_1$. We have developed a simplified kinetic model to describe the time dependence of the hyperpolarized signals observed in such

experiments, which include the chemical kinetics as well as the spin dynamics. This allows simultaneous fitting of the data from several experiments and estimation of most of the kinetic parameters and relaxation rate constants.

The particular hydrogenation reaction discussed here does not lead to a product molecule with biological function. Nevertheless, our results demonstrate the principle of methylene hyperpolarization by hydrogenative PHIP, and that the short $T_1$ values of these protons do not necessarily prevent the accumulation of hyperpolarization. We hope that this work might allow

exploration of a new range of hyperpolarized molecular targets.





## Appendix A: Full $^1$H NMR spectrum of reaction solution after hydrogenation

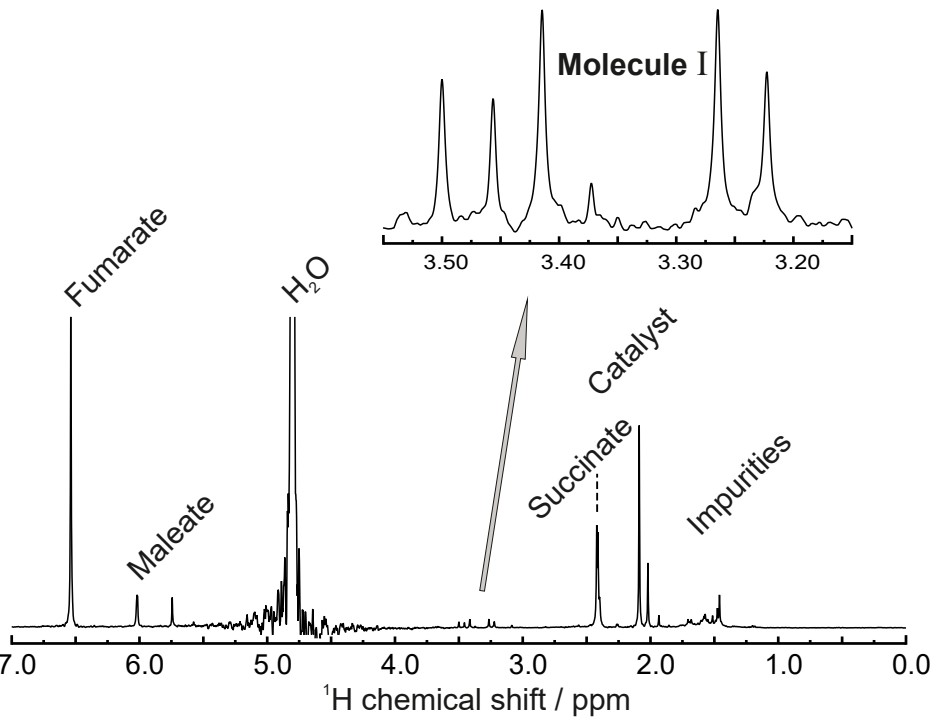

**Figure A1.** Full $^1$H NMR spectrum (400 MHz) of the reaction solution after hydrogenation. Apart from **I**, the solution contains several other substances, including fumarate, succinate and maleate.

## Appendix B: Asymmetry of partially-relaxed hyperpolarized spectra

The figure 2 shows hyperpolarized spectra after a PHIP experiment as well as spectra after relaxation period of 90 s. Although latter spectra represents almost fully relaxed spin system a minor asymmetry of the roof pattern can be seen. The explanation
for this observation is provided by simulation that is given below.

In the system of two coupled protons, the coherent Hamiltonian can be readily expressed in two terms of chemical shift and scalar coupling between two them:

$$\mathcal{H}_{\mathrm{coh}} = \pi\omega_\Delta(I_{1,\mathrm{z}} - I_{2,\mathrm{z}}) + 2\pi J I_1 I_2$$
$$\omega_\Delta = -\gamma_{\mathrm{H}}\mathbf{B}_0(\delta_1 - \delta_2) \tag{B1}$$

Note that Hamiltonian is expressed in rotating frame that is fixed to the mean resonance frequency of the two spins. Parameters for numerical simulation were set to $J = 16.9$ Hz and $\omega_\Delta/2\pi = 80$ Hz to recreate experimental observation.





Two different initial density matrices were considered. First one $\rho_1 = I_z$ simply corresponds to magnetization, where unity operator is neglected. The definition of the second density operator is slightly more complex. Parahydrogen during chemical reaction deposits its singlet order between protons which will evolve depending on Hamiltonian dictated by the new system.

Since the singlet order in the system is not an eigenoperator of the spin evolution, secular approximation to the initial singlet order needs to be applied. This omits any coherences created by instant projection of singlet order onto a new basis. Thus, the second density matrix ($\rho_2$) is a singlet state between two protons secularized according to the Hamiltonian with coupling and chemical shift values given above.

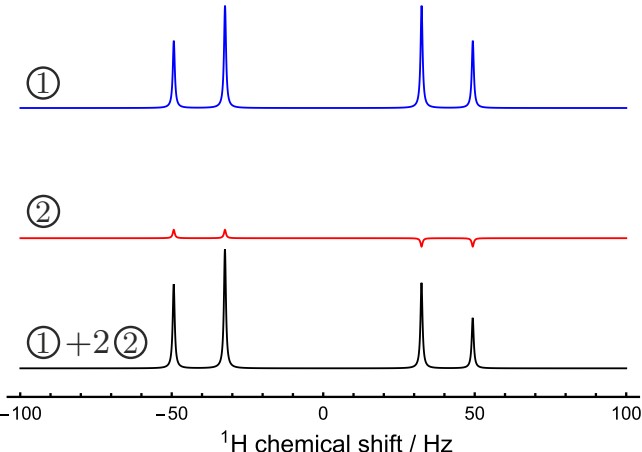

**Figure B1.** The origin of asymmetry observed in figure 2. Simulated spectra show signals of different density operators excited by the identical $\pi/2$ pulse. The first initial state is pure magnetization whereas the second one is singlet order secularized according to the Hamiltonian given by equation. B1

A strong $\pi/2$ pulse excites very different spectra for the two density operators. The first one possesses slight roofing effect

whereas the second one corresponds to so-called ALTADENA pattern. Now, if the sample contains fully relaxed spin ensemble with a similar amount of singlet order originating from previous parahydrogen reaction then the final spectra after $\pi/2$ pulse excitation will be a sum of two different patterns. This leads to the effect depicted in the figure B1. Notice that the magnitude of two density operators have to be comparable, meaning that singlet order is roughly equivalent to a thermal polarization level of protons which is very small in contrast to typical hyperpolarization yield in PHIP experiments.

*Author contributions.* Laurynas Dagys: Preparation, execution, and optimization of experiments. Data analysis, fitting of data. Support in kinetic model development. Manuscript writing.

Barbara Ripka: Preparation and execution of experiments. Project coordination. Manuscript writing.

Markus Leutzsch: Postulation of the chemical structure of **I** and the reaction path. Proposal for independent synthesis of **I** for its chemical structure determination. Consultation in the interpretation of NMR data. Contribution to manuscript writing.



Gamal A. I. Moustafa: Carried out the independent synthesis of **I** and determination of its chemical structure. Consultation in the interpretation of NMR spectra and in the identification of chemical degradation processes. Contribution to SI writing.

    James Eills: Performed first experiments, which indicated a hyperpolarized singlet state on the methylene protons of **I**. Consultation in experimental design and data fitting. Support in kinetic model development. Manuscript writing.

    Johannes F. P. Colell: Experimental setup engineering for finely adjustable and reproducible experiments. Consultation in experimental
design, interpretation of NMR spectra and identification of chemical degradation processes. Manuscript writing.

    Malcolm H. Levitt: Scientific supervision and idea generation. Consultation in NMR pulse sequence design, theoretical background, development of kinetic models, fitting of data, and manuscript writing.

*Competing interests.*  The authors declare that they have no conflict of interest.

*Acknowledgements.*  This project was funded by the Marie Skłodowska-Curie program of the European Union (grant number 766402),
the European Research Council (786707-FunMagResBeacons), EPSRC-UK (grants EP/P009980/1 and EP/P030491/1). Markus Leutzsch acknowledges generous support from the Max Planck Society.



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
