# Peer review of "Geminal Parahydrogen-Induced Polarization: Accumulating Long-Lived Singlet Order on Methylene Proton Pairs"

_Magnetic Resonance, 2020_

## Referee Comment (RC1) · Anonymous Referee #1 · 13 Jul 2020

The manuscript introduces an extension of hydrogenative PHIP allowing to hyperpolarize long-lived singlet states of CH2-groups and models the reaction kinetics and spin dynamics using a rate equation system. Even though in the system under study the polarization level turns out to be rather low, the method is intriguing and may lead to fascinating applications. Altogether, the manuscript is coherently written and is well suited for publication. There are a few points listed below that the authors should reconsider.

Page 5, line 87: here, "chemical instability" of target species I is shortly mentioned. A more specific information on the life-time is needed to ensure that this instability

doesn't interfere with the observed time evolution. Page 7, Figure 5: The first experimental points are given for tau sub2=5 s (with certain deviation from the model curve), while for shorter time only the model functions are shown. In my opinion, additional data measured at earlier time would make the agreement more compelling. At least for case c there is no reason to skip tau sub2=0. Also, the R value for the quality of the fit should be given. Page 17, Figure A1: The spectrum of molecule I does not agree with that shown in Figure 2. Chemical shift and number of lines are different. These differences should be commented. Also, the Appendices should be checked for language flaws and corrected accordingly.

Please also note the supplement to this comment:
https://mr.copernicus.org/preprints/mr-2020-16/mr-2020-16-RC1-supplement.pdf

**Supplement:**

**Reviewer Report**

**Manuscript Number:** mr-2020-16

**Title:** "Geminal Parahydrogen-Induced Polarization: Accumulating Long-Lived Singlet Order on Methylene Proton Pairs"

**Authors:** Laurynas Dagys, Barbara Ripka, Markus Leutzsch, Gamal A. I. Moustafa, James Eills, Johannes F. P. Colell, and Malcolm H. Levitt

The manuscript introduces an extension of hydrogenative PHIP allowing to hyperpolarize long-lived singlet states of $CH_2$-groups and models the reaction kinetics and spin dynamics using a rate equation system. Even though in the system under study the polarization level turns out to be rather low, the method is intriguing and may lead to fascinating applications. Altogether, the manuscript is coherently written and is well suited for publication. There are a few points listed below that the authors should reconsider.

Page 5, line 87: here, "chemical instability" of target species **I** is shortly mentioned. A more specific information on the life-time is needed to ensure that this instability doesn't interfere with the observed time evolution.

Page 7, Figure 5: The first experimental points are given for $\tau_2$=5 s (with certain deviation from the model curve), while for shorter time only the model functions are shown. In my opinion, additional data measured at earlier time would make the agreement more compelling. At least for case c there is no reason to skip $\tau_2$=0. Also, the R value for the quality of the fit should be given.

Page 17, Figure A1: The spectrum of molecule **I** does not agree with that shown in Figure 2. Chemical shift and number of lines are different. These differences should be commented. Also, the Appendices should be checked for language flaws and corrected accordingly.

---

## Referee Comment (RC2) · Anonymous Referee #2 · 20 Jul 2020

Dagys et al. conducted an interesting NMR study of a ruthenium-catalyzed geminal hydrogenation reaction producing overpopulated singlet spin order in a methylene group. By intelligent design of the pulse sequences and pH2 bubbling intervals, the authors studied the kinetics of the buildup and decay of the singlet order and zz- spin order in the intermediate of the hydrogenation reaction. This allowed them to extract relevant chemical and relaxation parameters and, in general, presented an elaborated strategy for the analysis of the reaction intermediates for the future experiments utilizing pH2. The work is well conducted, and the results are presented clearly. The paper without a doubt deserves publication in the Open Magnetic Resonance. The only puzzling fact is more than a factor of 2 difference between Ts measured at room temperature and at

PHIP conditions. The authors mention temperature as a potential reason for discrepancy... Would it be possible to conduct a singlet state lifetime measurement with the synthesized molecule at elevated temperature to support this statement?

I also suggest citing the following review relevant in the context of the interplay between chemical kinetics and spin dynamics in experiments utilizing pH2:

K. V. Kovtunov, E. V. Pokochueva, O. G. Salnikov, S. F. Cousin, D. Kurzbach, B. Vuichoud, S. Jannin, E. Y. Chekmenev, B. M. Goodson, D. A. Barskiy, I. V. Koptyug, Chem. Asian J. 2018, 13, 1857.

Otherwise, the paper is ready for publication.

––––––––––––––––––––––––––––––––

---

## Author Response (AR1)

**Author's Response**

**Response to referee #1**

We would like to thank anonymous Referee #1 for the kind remarks, and we are happy to answer the  important comments given in the review.

*"Page 5, line 87: here, "chemical instability" of target species I is shortly mentioned. A more specific information on the life-time is needed to ensure that this instability doesn't interfere with the observed time evolution."*

The target species I slowly decompose at elevated temperatures losing one carboxylic group. As a result, the chemical inequivalence of the two methylene protons is lost and the sharp line caused by this can be seen in Figure A1 around 3.5 ppm. We have not carried out a complete measurement to assess the rate of such process but based on our experience it seems that it takes some tens of minutes in the experimental conditions set for PHIP experiments. The decomposition is negligible over the timescale of a single NMR experiment. However, it does interfere with the generation of a reproducible series of experiments. We have added details on the nature and timescale of the decomposition in the Supporting Information.

*"Page 7, Figure 5: The first experimental points are given for $\tau 2$=5 s (with certain deviation from the model curve), while for shorter time only the model functions are shown. In my opinion, additional data measured at earlier time would make the agreement more compelling. At least for case c there is no reason to skip $\tau 2$=0. Also, the R value for the quality of the fit should be given."*

Thank you for the suggestion. Probing short time points would be a very interesting experiment. However, there are several major difficulties which are hard to avoid. The setup with which the experiments were done consists of mechanically controlled valves which are actuated by a user. The time precision required for small changes in time points would be a hard thing to attain without full automation. An automatic setup is under construction but not yet ready for use.

Additionally, we would like to stress an interesting point about the spin dynamics which we imposed into our trajectories. The theoretical expressions involve assumptions about the projections of spin operators which do not apply strictly for short times. We have added explanations and clarifications in the revised text.

R values have now been provided for the fits. Thank you for bringing this to our attention

*"Page 17, Figure A1: The spectrum of molecule I does not agree with that shown in Figure 2. Chemical shift and number of lines are different. These differences should be commented. Also, the Appendices should be checked for language flaws and corrected accordingly."*

The additional peak in the spectrum at 3.5 ppm can be assigned to the product of the decomposition of the chemical species I. This has now been indicated in the text.

The discrepancies in the chemical shifts are due to the different temperatures at which the two sets of experimental data were taken. These temperatures are now indicated in the figure captions.

The language flaws in the appendices have been corrected.

**Response to referee #2**

We thank the referee for his/her appreciative comments.

The referee asks: " The only puzzling fact is more than a factor of 2 difference between Ts measured at room temperature and at PHIP conditions. The authors mention temperature as a potential reason for discrepancy...Would it be possible to conduct a singlet state lifetime measurement with the synthesized molecule at elevated temperature to support this statement?"

As commented in the manuscript, the singlet decay rate measurement at elevated temperature is made more difficult by the chemical instability of the compound. More details on the chemical instability and its time scale are given in the revised manuscript.

The referee also wishes for a reference to be included. We were not aware of this article and we have now included it.

Attached manuscript contains **coloured** areas where these changes have been made. Note that a complete section was added in the new Supporting Information.

[revised manuscript text omitted]

$$Q_{\text{zz}} = 2I_{1z}I_{2z}$$

$$\textbf{zz} = \langle Q_{\text{zz}} \rangle = \text{Tr}\{Q_{\text{zz}}^{\dagger}\rho\}$$

(9)

In the absence of a spin-locking field, and if there is a relatively large chemical shift difference between the coupled spins, the operator $Q_{\text{zz}}$ is a better approximation to an eigenoperator of the spin evolution propagator than the singlet order operator $Q_{\text{SO}}$. The relaxation of the system can be complex and multi-exponential in this case. Nevertheless, for the sake of simplicity, we assume here a single rate constant $R_{\text{zz}} = T_{\text{zz}}^{-1}$ for the zz-order in the absence of a spin-locking field. The time constant $T_{\text{zz}}$ is expected to be close to the ordinary spin-lattice relaxation time constant $T_1$. The decay of zz-order in the absence of a spin-locking field, with rate constant $R_{\text{zz}}$, is shown in figure 6 by the dashed arrow running upwards connecting the **zz** state of the reaction product **I** to the unpolarized state.

**4.2.3 Spin-locking OFF**

Suppose that the molecules of **I** are in a state of enhanced singlet order **SO**. This state is stable if a spin-locking field is continuously applied, and decays monotonically with the time constant $T_{\text{S}}$, However, if the spin-locking field is turned off, the chemical shift difference between the methylene protons leads to rapid singlet-triplet mixing. The zz-order operator $Q_{\text{zz}}$ is an approximate eigenoperator of the evolution in this case, instead of the singlet-order operator $Q_{\text{SO}}$. Hence, any singlet order **SO** which is present when the spin-locking field is turned off is projected onto the zz-order operator $Q_{\text{zz}}$. The remaining spin order corresponds to zero-quantum coherences which rapidly oscillate and decay. These additional components may be ignored to a good approximation, providing that the spin-locking field remains turned off for an interval long compared to the difference in chemical shift frequencies.

The zz-order created by this projection process is given by

$$\textbf{zz} = \frac{\text{Tr}\{Q_{\text{zz}}^{\dagger}Q_{\text{SO}}\}}{\text{Tr}\{Q_{\text{zz}}^{\dagger}Q_{\text{zz}}\}}\textbf{SO} = -\frac{2}{3}\textbf{
[revised manuscript text omitted]